# SARS-CoV-2 Virus-Like Particle Neutralizing Capacity in Blood Donors Depends on Serological Profile and Donor-Declared SARS-CoV-2 Vaccination History

Steven J. Drews,[a,b] Queenie Hu,[c] Reuben Samson,[c,d] Kento T. Abe,[c,d] Bhavisha Rathod,[c] Karen Colwill,[c] Anne-Claude Gingras,[c,d] Qi-Long Yi,[e,f] Sheila F. O'Brien[e,f]

aDepartment of Microbiology, Canadian Blood Services, Edmonton, Alberta, Canada
bDepartment of Laboratory Medicine and Pathology, University of Alberta, Edmonton, Alberta, Canada
cLunenfeld-Tanenbaum Research Institute at Mt. Sinai Hospital, Sinai Health, Toronto, Ontario, Canada
dDepartment of Molecular Genetics, University of Toronto, Toronto, Ontario, Canada
eEpidemiology and Surveillance, Canadian Blood Services, Ottawa, Ontario, Canada
fSchool of Epidemiology and Public Health, University of Ottawa, Ottawa, Ontario, Canada

**ABSTRACT** This study attempted to understand the levels of neutralizing titers and the breadth of antibody protection against wild-type and variant severe acute respiratory syndrome coronavirus 2 (SARS-CoV-2) in Canadian blood donors during the first 3 months of 2021. During this period, it is unlikely that many of the blood donors had received a second dose, since vaccine rollout had not yet ramped up, and less than 2% of the Canadian population had received a second dose of vaccine. A repeated cross-sectional design was used. A random cross-sectional sampling of all available Canadian Blood Services retention samples ($n = 1,500$/month) was drawn monthly for January, February, and March 2021. A tiered testing approach analyzed 4,500 Canadian blood donor specimens for potential evidence of a signal for anti-spike (anti-S), anti-receptor-binding domain (anti-RBD), and anti-nucleocapsid protein (anti-N). Specimens were stratified based on donor-declared vaccination history and then stratified on the presence or absence of anti-N as follows: (i) "vaccinated plus anti-N" ($n = 5$), (ii) "vaccinated and no anti-N" ($n = 20$), (iii) "unvaccinated plus anti-N" ($n = 20$), and (iv) "unvaccinated and no anti-N" ($n = 20$). Randomized specimens were then characterized for neutralizing capacity against wild-type as well as SARS-CoV-2 variants of concern (VOCs) (Alpha [B.1.1.7], Beta [B.1.351], Gamma [P.1], and Delta [B.1.617.2]) using S-pseudotyped virus-like particle (VLP) neutralization assays. There was no neutralizing capacity against wild-type and VOC VLPs within the "no vaccine and no anti-N" group. Neutralization of Beta VLPs was less than wild-type VLPs within "vaccinated plus anti-N," "vaccinated and no anti-N", and "unvaccinated plus anti-N" groups.

**IMPORTANCE** In the first 3 months of 2021 as severe acute respiratory syndrome coronavirus 2 (SARS-CoV-2) vaccination was in the initial stages of a mass rollout, Canadian blood donors had various levels of humoral protection against wild-type and variant of concern (VOC) SARS-CoV-2. Very few Canadians would have received a second dose of a SARS-CoV-2 vaccine. In this study, we identified elevated levels of neutralizing capacity, albeit with reduced neutralization capacity against one or more SARS-CoV-2 strains (wild type and VOCs) in vaccinated blood donors. This broad neutralizing response we present regardless of evidence of natural SARS-CoV-2 infection. Neutralizing capacity against wild type and VOCs varied significantly within the unvaccinated group, with one subset of unvaccinated plasma specimens (unvaccinated and no anti-N) having no measurable wild type- nor variant-neutralizing capacity. The study is important because it indicates that vaccination can be

Address correspondence to Steven J. Drews, steven.drews@blood.ca.

The authors declare a conflict of interest. The following authors have no conflicts of interest: S.F.O., Q.H., R.S., K.T.A., B.R., K.C., and Q.-L.Y. S.J.D. has functioned as a content expert for respiratory viruses for Johnson & Johnson (Janssen). A.-C.G. receives research funding from Providence Therapeutics Holdings, Inc. for other projects.

associated with a broad neutralizing antibody capacity of donor plasma against SARS-CoV-2 VOCs.

**KEYWORDS** SARS-CoV-2 antibody, neutralizing antibodies, nucleocapsid, receptor-binding domain, spike, variants of concern, virus-like particles

Since the first identification of severe respiratory syndrome coronavirus 2 (SARS-CoV-2) virus in Canada, there have been four waves of activity, approximately 1.65 million cases, and 28,000 deaths due to coronavirus disease 2019 (COVID-19) (1). A first wave occurred between late February and early July of 2020 and was followed by a higher-amplitude second wave that inflected in early August, peaked in late December, and ebbed by mid-March 2021. A third wave started in late March 2021 and was active until late June 2021. The fourth wave soon followed and was active as of November 2021 (1). Prior seroprevalence surveys of Canadian blood donors by Canadian Blood Services suggested that donors had very low levels of seropositivity (e.g., <5 %) between April 2020 and the second week of January 2021 ($n = 172,919$) (2, 3).

By March 2020, Canadian Blood Services began to collaborate with several laboratories in Canada and the United States with the goal of characterizing the neutralizing capacity of anti-SARS-CoV-2 antibodies in Canadian blood donors (4–7). This work supported both Canadian convalescent plasma clinical trials as well as national seroprevalence work by the Canadian COVID-19 Immunity Task Force (CITF) (1, 6, 8–11). A variety of serological detection and neutralization assays were utilized between April 2020 and March of 2021. From these experiments, it was clear that prior to SARS-CoV-2 vaccine rollout in Canada, a strong sustained anti-SARS-CoV-2-neutralizing capacity was not evident in all blood donors and/or convalescent plasma donors with serological evidence of a past SARS-CoV-2 infection (3, 6, 7).

The SARS-CoV-2 vaccine was first administered in Canada on 14 December 2020 (12). As of 20 March 2021, 3,487,915 people (9.18% of the population) had received at least one dose of a SARS-CoV-2 vaccine (1 dose: 2,857,576 people [7.52% of the population]). A minute percentage of the population received two doses by that time (630,339 people [1.66% of the population]). From January to March 2021, it is unlikely that many of the blood donors had received a second dose, since vaccine rollout had not yet ramped up, and less than 2% of the Canadian population had received a second dose of vaccine (13). Vaccine administration was skewed to mRNA vaccines, with 6.69% of the population receiving at least one dose of BNT162b2 (Pfizer-BioNTech, New York, NY, USA; 5.49% received one dose, and 1.20% received two doses). Another 1.66% of the population received at least one dose of mRNA-1273 (Moderna, Cambridge, MA, USA; 1.21% received one dose, and 0.45% received two doses). Finally, 0.81% of the population received one dose of ChAdOx1-nCOV (Serum Institute of India: Covishield, Pune, India, licensed from AstraZeneca, Cambridge, UK); no one received two doses of that vaccine. In that time period, no one in Canada had received Ad26.COV2.S (Janssen, Raritan, NJ, USA) (13). Vaccination targeted health care workers, adults aged 80 years of age or older, and individuals living in senior citizen group care settings. SARS-CoV-2 vaccine supplies were insufficient in this time period, and Canada would later move to extended dosing intervals for Health Canada-approved vaccines (14). A survey of Canadian blood donors from 24 March 2021 indicated that 67% of the vaccinated donors were essential or frontline workers (S. O'Brien, unpublished data).

Given differences in national definitions for variants of concern (VOCs) and variants of interest, the manuscript will use the term "variant(s) of concern" to describe Alpha (B.1.1.7), Beta (B.1.351), Gamma (P.1), and Delta (B.1.617.2) (15, 16). The landscape of circulating VOCs during the first 3 months of 2021 was diverse and changing. By late March 2021, it was estimated that two-thirds of COVID-19 cases in the Canadian province of Ontario were due to VOCs (17). A survey of SARS-CoV-2 in Ontario indicated that there was an increasing proportion of N501Y VOCs (e.g., Alpha, Beta, and Gamma) identified between mid-February and late March 2021 (18). This was a very different epidemiologic landscape compared to September 2021, when most SARS-CoV-2-

**TABLE 1** Canadian Blood Services blood donor demographics for the period between January 2021 and March 2021

| Demographic | All | Vaccinated (n [%]) | Unvaccinated (n [%]) |
|---|---|---|---|
| N | 4,500 | 138 (3.1) | 4,362 (96.9) |
| Female | 2,035 | 92 (66.7) | 1,943 (45.5) |
| Male | 2,465 | 46 (33.3) | 2,419 (55.5) |
| Median age (range), yrs | 47 (17–85) | 39 (21–85) | 47 (17–84) |
| | | | |
| Region | | | |
| Western Canada | 2,085 | 56 (2.7) | 2,029 (97.3) |
| Alberta | 922 | 15 (1.6) | 907 (98.4) |
| British Columbia | 682 | 21 (3.1) | 661 (96.9) |
| Manitoba | 240 | 14 (5.8) | 226 (94.2) |
| Saskatchewan | 241 | 6 (2.5) | 235 (97.5) |
| Eastern Canada and other | 2,415 | 81 (3.4) | 2,334 (96.6) |
| New Brunswick | 153 | 6 (3.9) | 147 (96.1) |
| Newfoundland and Labrador | 81 | 2 (2.5) | 79 (97.5) |
| Nova Scotia | 205 | 5 (2.4) | 200 (97.5) |
| Ontario | 1,934 | 66 (3.4) | 1,868 (96.5) |
| Prince Edward Island | 34 | 2 (5.9) | 32 (94.1) |
| Other[a] | 8 | 0 (0) | 8 (100) |

[a]Other includes Quebec ($n = 6$), United States ($n = 1$), and no information ($n = 1$).

positive specimens (98.9%) in Canada were Delta VOCs (17). Publicly available data on VOCs (17) may represent a sampling of convenience, which could be biased by variability in provincial surveillance and sequencing strategies.

During the first 3 months of 2021, it was clear that multiple variables may have been impacting the humoral immunity of Canadian blood donors against wild-type and variant SARS-CoV-2. To understand these variables, this study was undertaken to characterize the neutralizing capacity of plasma from Canadian blood donors against wild-type and variant SARS-CoV-2 from this period.

## RESULTS

**Blood donor demographics.** In total, 4,500 specimens were analyzed between 1 January and 31 March 2021 (Table 1). Sex distribution slightly favored males ($n = 2,465$, 54.8%) versus females ($n = 2,035$, 45.2%). The median age of all donors was 47 years of age (range of 17 to 85 years of age).

We were unable to capture information on vaccine type and if a second dose of vaccine had been administered. We were also only able to assesses donors for the receipt of a SARS-CoV-2 vaccine in the prior 3 months and not the timing of the vaccine. In this period, a minority of Canadian blood donors declared that they were vaccinated with at least one dose of a SARS-CoV-2 vaccine in the past 3 months ($n = 138$, 3.1%), while most remained unvaccinated ($n = 4,362$, 96.9%). Vaccinated donors (median of 39 years of age, range of 21 to 85 years of age) were younger than unvaccinated donors (median of 47 years of age, range of 17 to 84 years of age) (Mann-Whitney U test, two tailed; $P < 0.0001$) (Table 1). Vaccinated donors were also more likely to be female ($n = 92/138$, 66.7%), while unvaccinated donors were slightly more likely to be male ($n = 2,419/4,362$, 55.5%) (Fisher's exact test, two sided; $P < 0.0001$) (Table 1).

**Enzyme immunoassay screening of specimens.** For this study, 1,500 specimens each from January, February, and March of 2021 (total: $n = 4,500$) were analyzed with previously published assays (5, 6, 19) for SARS-CoV-2 antigens: anti-Spike (S), anti-receptor-binding domain (RBD), and anti-nucleocapsid (N) (Sinai Health anti-N and Abbott anti-N) (Fig. 1). Signal-to-cutoff (S/Co) distributions for anti-S, anti-RBD, anti-N (Sinai Health), and Abbott anti-N are shown for all vaccinated (Fig. 2A and C) and all unvaccinated (Fig. 2B and D) donors in this study ($n = 4,500$).

**Selection of SARS-CoV-2 seroreactive specimens for virus-like particle neutralization.** As in Fig. 1, all anti-S and/or anti-RBD specimens ($n = 320$) (7.1%) of 4,500 specimens were first stratified based on donor-declared vaccination history (dosing number and vaccine type information was not available but was expected to be mostly one dose) and then stratified

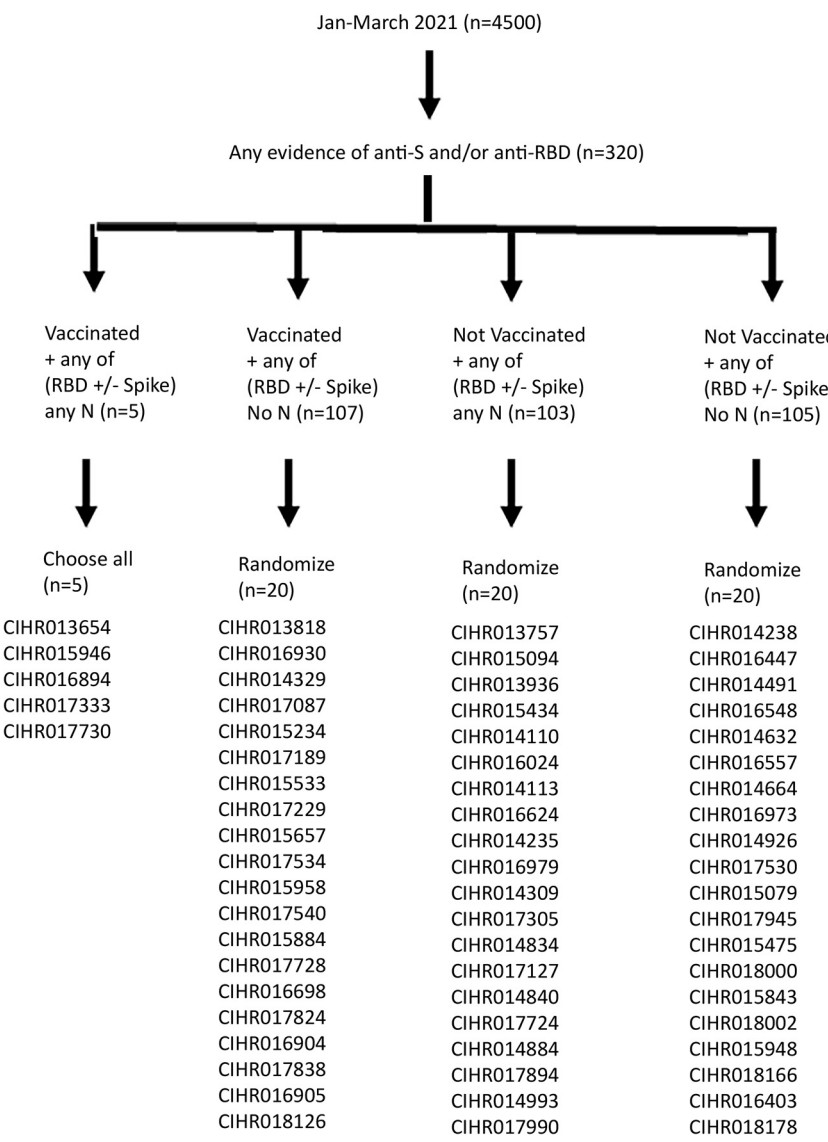

**FIG 1** A tiered approach to identify specimens sent for wild-type and variant SARS-CoV-2 VLP neutralization testing. From January to March 2021, 4,500 specimens were first evaluated by SARS-CoV-2 enzyme immunoassays. Specimens were stratified based on donor-declared vaccination history. Specimens were also stratified based on positivity for anti-S, anti-RBD, and anti-N. All specimens (*n* = 5) from vaccinated donors with any anti-RBD and/or anti-S and anti-N were characterized by SARS-CoV-2 VLP assays. Specimens were scored as anti-N if they were positive by either the Sinai Health anti-N or the Abbott anti-N. For the remaining groups, approximately 20% of specimens were chosen by randomization for further SARS-CoV-2 VLP neutralization.

based on the presence or absence of anti-N. Specimens were categorized as anti-N positive if they were positive by either anti-N used in this study (Sinai Health anti-N or Abbott anti-N). Distributions of all S/Co values for anti-S, anti-RBD, anti-N (Sinai Health), and Abbott anti-N in specimens selected for virus-like particles (VLPs) are shown in Fig. 2A to D.

Due to sparse numbers, all specimens (*n* = 5) from vaccinated donors with anti-N positivity were selected for neutralization assessment with a VLP neutralization assay (5, 20). For the three other groups (ranging from 103 to 105 specimens per group), randomization was used to select 20 samples per group (i.e., 18.7 to 19.4% of the samples) to test for neutralization. Canadian Institutes of Health Research (CIHR) study numbers assigned to each group are identified in Fig. 1 and will be used throughout the manuscript. Individual immunoassay results for each of the 65 specimens are listed in Table 2.

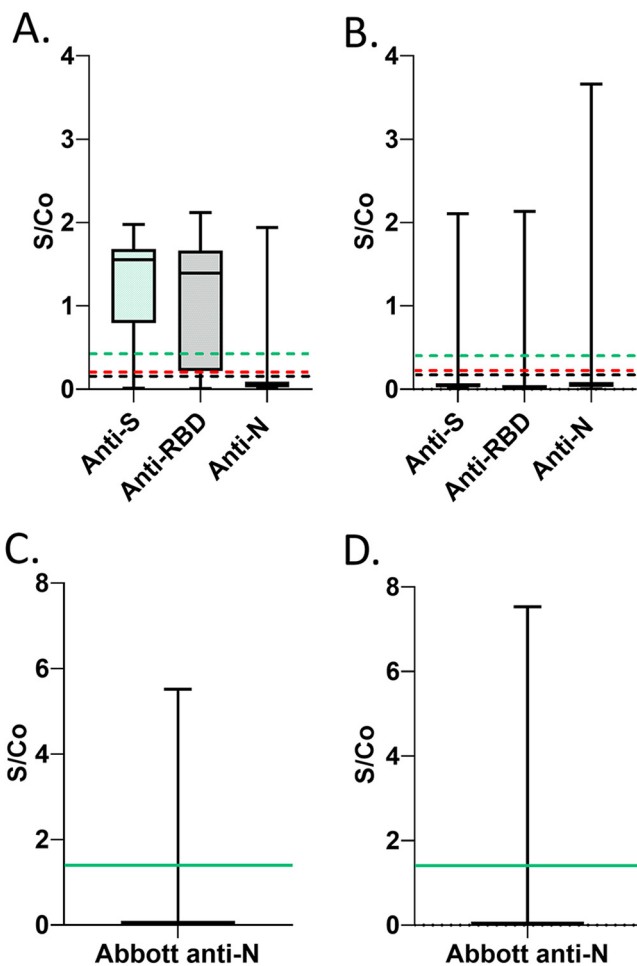

**FIG 2** Distribution of signal-to-cutoff values for anti-S, anti-RBD, anti-N, and Abbott anti-N in all vaccinated and unvaccinated blood donor specimens ($n$ = 4,500). Data are represented by box and whisker plots showing median, minimum, and maximum values. Each box extends from the 25th to 75th percentiles. Sinai Health and Abbott anti-N assay S/Co results are presented in different frames due to differences in dynamic range of S/Co. Assay cutoffs are drawn on graphs as anti-RBD (dashed black line, cutoff of 0.186), anti-S (dashed red line, cutoff of 0.190), anti-N (Sinai Health assay, dashed green line, cutoff of 0.396), and Abbott anti-N (solid green line, cutoff of 1.4). (A) Vaccinated donor specimens ($n$ = 138) using the Sinai Health assays. (B) Unvaccinated donor specimens using the Sinai Health assays ($n$ = 4,362). (C) Vaccinated donor specimens ($n$ = 138) using the Abbott anti-N assay. (D) Unvaccinated donor specimens using the Abbott anti-N assay ($n$ = 4,362). Ratio-converted ELISA reads were undertaken as previously described (5, 20), and cutoffs (positive) for each of the targets were ≥0.396 for anti-N (Sinai Health), ≥0.186 for anti-RBD, and ≥0.190 for anti-S (2). The cutoff for the Abbott anti-N was ≥1.40 (2).

For the selected subset of specimens, distributions of S/Co values for anti-S, anti-RBD, anti-N (Sinai Health), and Abbott anti-N in specimens selected for VLPs are shown in Fig. 3A to D ($n$ = 65). As in Fig. 3A, S/Co values for anti-S in the vaccinated group were higher in the vaccinated group (median 1.6) than in the unvaccinated group (median 0.46; Mann-Whitney $U$ = 131; $P$ < 0.0001, two tailed, exact). S/Co values for anti-RBD in the vaccinated group were higher in the vaccinated group (median 1.5) than in the unvaccinated group (median 0.18; Mann-Whitney $U$ = 133; $P$ < 0.0001, two tailed, exact) (Fig. 3B). S/Co values for anti-N (Sinai Health) in the vaccinated group (median 0.050) were lower than in the unvaccinated group (median 0.39; Mann-Whitney $U$ = 268; $P$ = 0.0014, two tailed, exact) (Fig. 3C). S/Co values for anti-N (Abbott) were lower in the vaccinated group (median 0.030) than in the unvaccinated group (median 0.59; Mann-Whitney $U$ = 285; $P$ = 0.0031, two tailed, exact) (Fig. 3D).

**Differences in VLP neutralization between wild-type, Alpha, Beta, Gamma, and Delta VLPs within vaccinated and unvaccinated groups.** Untransformed S-pseudo-typed virus-like particle (VLP) neutralization assay results for the wild-type parental

**TABLE 2** Summary of immunoassay results in a subset of specimens chosen for VLP neutralization in vaccinated and unvaccinated blood donors (January to March 2021)

| Sample ID | Vaccinated | Anti-S ratio | Anti-S[a] | Anti-RBD ratio | Anti-RBD[a] | Anti-N (Sinai Health) ratio | Anti-N (Sinai Health)[a] | Abbott anti-N ratio | Abbott anti-N[a] |
|---|---|---|---|---|---|---|---|---|---|
| CIHR013654 | Yes | 1.69 | Pos | 1.41 | Pos | 0.41 | Pos | 0.02 | Neg |
| CIHR013818 | Yes | 1.08 | Pos | 0.77 | Pos | 0.03 | Neg | 0.01 | Neg |
| CIHR014329 | Yes | 1.86 | Pos | 1.88 | Pos | 0.03 | Neg | 0.02 | Neg |
| CIHR015234 | Yes | 1.48 | Pos | 0.75 | Pos | 0.09 | Neg | 0.01 | Neg |
| CIHR015533 | Yes | 1.64 | Pos | 1.81 | Pos | 0.04 | Neg | 0.09 | Neg |
| CIHR015657 | Yes | 1.64 | Pos | 1.60 | Pos | 0.06 | Neg | 0.09 | Neg |
| CIHR015884 | Yes | 1.88 | Pos | 1.90 | Pos | 0.34 | Neg | 0.13 | Neg |
| CIHR015946 | Yes | 1.24 | Pos | 0.61 | Pos | 0.60 | Pos | 0.03 | Neg |
| CIHR015958 | Yes | 1.77 | Pos | 1.57 | Pos | 0.02 | Neg | 0.03 | Neg |
| CIHR016698 | Yes | 1.74 | Pos | 1.76 | Pos | 0.01 | Neg | 0.04 | Neg |
| CIHR016894 | Yes | 1.49 | Pos | 1.59 | Pos | 1.60 | Pos | 5.52 | Pos |
| CIHR016904 | Yes | 1.13 | Pos | 0.13 | Neg | 0.02 | Neg | 0.01 | Neg |
| CIHR016905 | Yes | 1.48 | Pos | 1.51 | Pos | 0.01 | Neg | 0.04 | Neg |
| CIHR016930 | Yes | 0.04 | Neg | 0.25 | Pos | 0.05 | Neg | 0.01 | Neg |
| CIHR017087 | Yes | 1.61 | Pos | 1.49 | Pos | 0.01 | Neg | 0.03 | Neg |
| CIHR017189 | Yes | 1.58 | Pos | 1.41 | Pos | 0.05 | Neg | 0.03 | Neg |
| CIHR017229 | Yes | 0.31 | Pos | 0.09 | Neg | 0.03 | Neg | 0.01 | Neg |
| CIHR017333 | Yes | 1.69 | Pos | 1.71 | Pos | 0.40 | Pos | 0.03 | Neg |
| CIHR017534 | Yes | 0.93 | Pos | 0.31 | Pos | 0.03 | Neg | 0.03 | Neg |
| CIHR017540 | Yes | 1.53 | Pos | 1.37 | Pos | 0.05 | Neg | 0.03 | Neg |
| CIHR017728 | Yes | 1.82 | Pos | 1.83 | Pos | 0.11 | Neg | 0.04 | Neg |
| CIHR017730 | Yes | 1.65 | Pos | 1.73 | Pos | 1.94 | Pos | 5.31 | Pos |
| CIHR017824 | Yes | 1.50 | Pos | 1.61 | Pos | 0.02 | Neg | 0.03 | Neg |
| CIHR017838 | Yes | 1.53 | Pos | 1.53 | Pos | 0.02 | Neg | 0.05 | Neg |
| CIHR018126 | Yes | 1.70 | Pos | 1.50 | Pos | 0.07 | Neg | 0.06 | Neg |
| CIHR013757 | No | 0.67 | Pos | 0.33 | Pos | 0.53 | Pos | 1.13 | Neg |
| CIHR013936 | No | 1.43 | Pos | 0.67 | Pos | 0.43 | Pos | 1.32 | Neg |
| CIHR014110 | No | 0.68 | Pos | 0.15 | Neg | 1.60 | Pos | 2.74 | Pos |
| CIHR014113 | No | 1.49 | Pos | 0.84 | Pos | 1.40 | Pos | 3.08 | Pos |
| CIHR014235 | No | 1.44 | Pos | 1.14 | Pos | 1.01 | Pos | 2.76 | Pos |
| CIHR014238 | No | 0.28 | Pos | 0.04 | Neg | 0.02 | Neg | 0.03 | Neg |
| CIHR014309 | No | 1.75 | Pos | 1.66 | Pos | 1.86 | Pos | 4.59 | Pos |
| CIHR014491 | No | 0.25 | Pos | 0.06 | Neg | 0.05 | Neg | 0.02 | Neg |
| CIHR014632 | No | 0.42 | Pos | 0.10 | Neg | 0.10 | Neg | 0.3 | Neg |
| CIHR014664 | No | 0.20 | Pos | 0.02 | Neg | 0.09 | Neg | 0.03 | Neg |
| CIHR014834 | No | 1.39 | Pos | 1.23 | Pos | 2.03 | Pos | 6.45 | Pos |
| CIHR014840 | No | 1.21 | Pos | 1.82 | Pos | 0.87 | Pos | 2.32 | Pos |
| CIHR014884 | No | 1.21 | Pos | 0.84 | Pos | 0.52 | Pos | 2.56 | Pos |
| CIHR014926 | No | 0.52 | Pos | 0.04 | Neg | 0.35 | Neg | 0.01 | Neg |
| CIHR014993 | No | 0.62 | Pos | 0.17 | Neg | 1.84 | Pos | 0.97 | Neg |
| CIHR015079 | No | 0.19 | Pos | 0.07 | Neg | 0.17 | Neg | 0.06 | Neg |
| CIHR015094 | No | 1.34 | Pos | 1.11 | Pos | 1.34 | Pos | 5.17 | Pos |
| CIHR015434 | No | 0.36 | Pos | 0.36 | Pos | 1.17 | Pos | 3.09 | Pos |
| CIHR015475 | No | 0.27 | Pos | 0.03 | Neg | 0.09 | Neg | 0.01 | Neg |
| CIHR015843 | No | 0.31 | Pos | 0.02 | Neg | 0.02 | Neg | 0.01 | Neg |
| CIHR015948 | No | 0.30 | Pos | 0.02 | Neg | 0.05 | Neg | 0.17 | Neg |
| CIHR016024 | No | 1.05 | Pos | 0.42 | Pos | 0.50 | Pos | 2.23 | Pos |
| CIHR016403 | No | 0.22 | Pos | 0.06 | Neg | 0.13 | Neg | 0.02 | Neg |
| CIHR016447 | No | 0.18 | No | 0.19 | Pos | 0.30 | Neg | 0.06 | Neg |
| CIHR016548 | No | 0.47 | Pos | 0.11 | Neg | 0.16 | Neg | 0.05 | Neg |
| CIHR016557 | No | 0.22 | Pos | 0.02 | Neg | 0.04 | Neg | 0.03 | Neg |
| CIHR016624 | No | 1.17 | Pos | 0.56 | Pos | 1.26 | Pos | 2.82 | Pos |
| CIHR016973 | No | 0.27 | Pos | 0.01 | Neg | 0.07 | Neg | 0.3 | Neg |
| CIHR016979 | No | 0.79 | Pos | 0.24 | Pos | 0.53 | Pos | 1.66 | Pos |
| CIHR017127 | No | 0.33 | Pos | 0.40 | Pos | 0.55 | Pos | 0.06 | Neg |
| CIHR017305 | No | 1.11 | Pos | 0.67 | Pos | 2.09 | Pos | 6.95 | Pos |
| CIHR017530 | No | 0.46 | Pos | 0.01 | Neg | 0.04 | Neg | 0.1 | Neg |
| CIHR017724 | No | 0.97 | Pos | 0.51 | Pos | 0.82 | Pos | 1.96 | Pos |
| CIHR017894 | No | 1.55 | Pos | 1.14 | Pos | 0.52 | Pos | 3.1 | Pos |
| CIHR017945 | No | 0.19 | Pos | 0.01 | Neg | 0.01 | Neg | 0.02 | Neg |

**TABLE 2** (Continued)

| Sample ID | Vaccinated | Anti-S ratio | Anti-S[a] | Anti-RBD ratio | Anti-RBD[a] | Anti-N (Sinai Health) ratio | Anti-N (Sinai Health)[a] | Abbott anti-N ratio | Abbott anti-N[a] |
|---|---|---|---|---|---|---|---|---|---|
| CIHR017990 | No | 1.00 | Pos | 0.37 | Pos | 0.97 | Pos | 2.61 | Pos |
| CIHR018000 | No | 0.26 | Pos | 0.12 | Neg | 0.31 | Neg | 0.88 | Neg |
| CIHR018002 | No | 0.33 | Pos | 0.19 | Pos | 0.08 | Neg | 0.03 | Neg |
| CIHR018166 | No | 0.25 | Pos | 0.01 | Neg | 0.02 | Neg | 0.01 | Neg |
| CIHR018178 | No | 0.22 | Pos | 0.01 | Neg | 0.02 | Neg | 0.03 | Neg |

[a]Pos, positive; Neg, negative.

strain (Wuhan-Hu-1 sequence) and Alpha, Beta, Gamma, and Delta VOCs are listed for each specimen in Table 3 (see transformed data in Tables S1 and S2 in the supplemental material). As these would be repeated measures within groups, Friedman statistics were used.

Except for the "unvaccinated and no anti-N" group (Fig. S1A to D), all other groups displayed measurable neutralization activity against wild-type and VOC VLPs (Fig. 4A to C; Table S2).

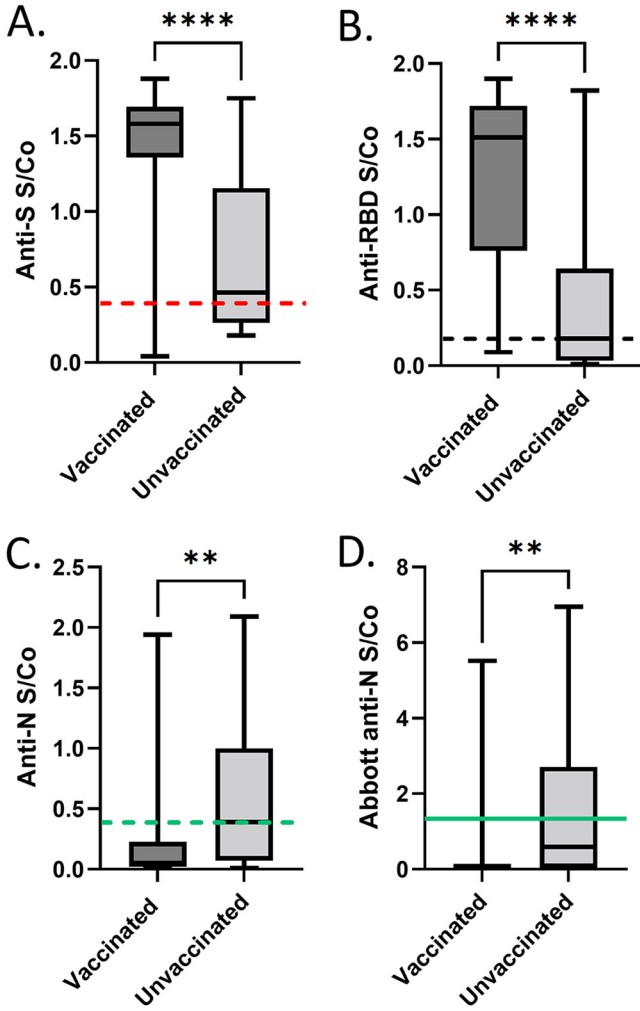

**FIG 3** Distribution of signal-to-cutoff values for anti-S, anti-RBD, anti-N, and Abbott anti-N in vaccinated and unvaccinated blood donor specimens chosen for VLP analysis (*n* = 65). Data are represented by box and whisker plots showing median, minimum, and maximum values. Each box extends from the 25th to 75th percentiles (vaccinated, *n* = 25; unvaccinated, *n* = 40). Assay cutoffs are drawn on graphs as anti-RBD (dashed black line, cutoff of 0.186), anti-S (dashed red line, cutoff of 0.190), anti-N (Sinai Health assay, dashed green line, cutoff of 0.396), and Abbott anti-N (solid green line, cutoff of 1.4). (A) Anti-S. (B) Anti-RBD. (C) Anti-N (Sinai Health). (D) Abbott anti-N. Ratio-converted ELISA reads undertaken for each of the targets were ≥0.396 for anti-N (Sinai Health), ≥0.186 for anti-RBD, and ≥0.190 for anti-S. The cutoff for the Abbott anti-N was ≥1.40 (2). Statistical comparisons were done by Mann-Whitney U analysis. Asterisks (*) denote significance.

**TABLE 3** Summary of untransformed VLP neutralization in vaccinated and unvaccinated blood donors (January to March 2021) against wild-type and VOC SARS-CoV-2

| Specimen no. | Vaccinated | Anti-N | $ID_{50}$ Wild-type | Alpha | Beta | Gamma | Delta |
|---|---|---|---|---|---|---|---|
| **Vaccinated + any of (RBD ± S) any N** | | | | | | | |
| CIHR013654 | Yes | Yes | $1.51 \times 10^2$ | $3.36 \times 10^1$ | 1.00 | 1.00 | $1.08 \times 10^1$ |
| CIHR015946 | Yes | Yes | $3.96 \times 10^1$ | 1.00 | 1.00 | 1.00 | 1.00 |
| CIHR016894 | Yes | Yes | $5.75 \times 10^4$ | $2.55 \times 10^4$ | $1.20 \times 10^4$ | $2.65 \times 10^4$ | $3.02 \times 10^4$ |
| CIHR017333 | Yes | Yes | $4.45 \times 10^3$ | $2.37 \times 10^3$ | $2.81 \times 10^2$ | $1.70 \times 10^3$ | $2.97 \times 10^3$ |
| CIHR017730 | Yes | Yes | $1.62 \times 10^4$ | $7.26 \times 10^3$ | $4.00 \times 10^3$ | $5.35 \times 10^3$ | $6.24 \times 10^3$ |
| **Vaccinated + any of (RBD ± S) no N** | | | | | | | |
| CIHR013818 | Yes | No | $2.13 \times 10^3$ | $1.54 \times 10^3$ | 1.00 | 1.00 | $2.65 \times 10^3$ |
| CIHR014329 | Yes | No | $2.45 \times 10^2$ | $1.12 \times 10^2$ | $2.74 \times 10^1$ | $9.17 \times 10^1$ | $2.22 \times 10^2$ |
| CIHR015234 | Yes | No | $1.07 \times 10^2$ | 1.00 | 1.00 | 1.00 | 1.00 |
| CIHR015533 | Yes | No | $1.83 \times 10^4$ | $6.41 \times 10^3$ | $6.81 \times 10^2$ | $3.17 \times 10^3$ | $5.14 \times 10^3$ |
| CIHR015657 | Yes | No | $3.97 \times 10^2$ | $1.11 \times 10^2$ | $38.9 \times 10^1$ | $1.24 \times 10^2$ | $1.51 \times 10^2$ |
| CIHR015884 | Yes | No | $1.98 \times 10^3$ | $7.16 \times 10^2$ | $2.07 \times 10^2$ | $9.97 \times 10^2$ | $1.28 \times 10^3$ |
| CIHR015958 | Yes | No | $1.12 \times 10^3$ | $1.85 \times 10^2$ | 1.00 | $7.90 \times 10^1$ | $1.90 \times 10^2$ |
| CIHR016698 | Yes | No | $1.48 \times 10^3$ | $1.01 \times 10^3$ | $1.23 \times 10^2$ | $7.84 \times 10^2$ | $7.07 \times 10^2$ |
| CIHR016904 | Yes | No | $8.13 \times 10^1$ | $2.17 \times 10^1$ | 1.00 | 1.00 | $5.39 \times 10^1$ |
| CIHR016905 | Yes | No | $4.70 \times 10^3$ | $2.42 \times 10^3$ | $4.99 \times 10^2$ | $9.92 \times 10^2$ | $2.39 \times 10^3$ |
| CIHR016930 | Yes | No | 1.00 | 1.00 | 1.00 | 1.00 | 1.00 |
| CIHR017087 | Yes | No | $3.59 \times 10^2$ | $1.20 \times 10^2$ | $7.70 \times 10^1$ | $3.14 \times 10^2$ | $4.06 \times 10^2$ |
| CIHR017189 | Yes | No | $1.18 \times 10^3$ | $4.24 \times 10^2$ | $9.15 \times 10^1$ | $6.30 \times 10^2$ | $5.13 \times 10^2$ |
| CIHR017229 | Yes | No | 1.00 | 1.00 | 1.00 | 1.00 | 1.00 |
| CIHR017534 | Yes | No | $5.53 \times 10^1$ | $4.89 \times 10^1$ | 1.00 | 1.00 | 1.00 |
| CIHR017540 | Yes | No | $6.69 \times 10^2$ | $1.41 \times 10^2$ | 1.00 | $1.02 \times 10^2$ | $7.45 \times 10^1$ |
| CIHR017728 | Yes | No | $1.13 \times 10^4$ | $4.73 \times 10^3$ | $8.37 \times 10^2$ | $2.37 \times 103$ | $2.67 \times 10^3$ |
| CIHR017824 | Yes | No | $1.40 \times 10^3$ | $4.30 \times 10^2$ | $2.65 \times 10^2$ | $6.54 \times 10^2$ | $3.09 \times 10^2$ |
| CIHR017838 | Yes | No | $9.35 \times 10^3$ | $2.79 \times 10^3$ | $6.21 \times 10^2$ | $1.82 \times 10^3$ | $2.02 \times 10^3$ |
| CIHR018126 | Yes | No | $7.97 \times 10^2$ | $5.62 \times 10^2$ | $1.15 \times 10^2$ | $5.17 \times 10^2$ | $3.09 \times 10^2$ |
| **Unvaccinated + any of (RBD ± S) any N** | | | | | | | |
| CIHR013757 | No | Yes | 1.00 | 1.00 | 1.00 | 1.00 | 1.00 |
| CIHR013936 | No | Yes | $4.91 \times 10^2$ | $1.93 \times 10^2$ | $8.09 \times 10^1$ | $1.44 \times 10^2$ | $1.55 \times 10^2$ |
| CIHR014110 | No | Yes | $7.02 \times 10^2$ | $1.25 \times 10^2$ | $2.29 \times 10^2$ | $3.94 \times 10^2$ | $1.24 \times 10^2$ |
| CIHR014113 | No | Yes | $1.27 \times 10^3$ | $1.39 \times 10^2$ | $5.06 \times 10^1$ | $3.03 \times 10^2$ | $1.41 \times 10^2$ |
| CIHR014235 | No | Yes | $1.01 \times 10^3$ | $3.76 \times 10^2$ | $3.74 \times 10^1$ | $6.23 \times 10^1$ | $9.52 \times 10^2$ |
| CIHR014309 | No | Yes | $3.39 \times 10^2$ | $1.28 \times 10^2$ | $5.06 \times 10^1$ | $1.88 \times 10^2$ | $1.3 \times 10^2$ |
| CIHR014834 | No | Yes | $6.63 \times 10^1$ | $4.13 \times 10^1$ | 1.00 | 1.00 | $3.91 \times 10^1$ |
| CIHR014840 | No | Yes | $1.85 \times 10^2$ | $9.53 \times 10^1$ | $2.33 \times 10^1$ | $4.38 \times 10^1$ | $5.43 \times 10^1$ |
| CIHR014884 | No | Yes | $5.65 \times 10^2$ | $1.21 \times 10^2$ | $3.12 \times 10^1$ | $3.29 \times 10^2$ | $1.22 \times 10^2$ |
| CIHR014993 | No | Yes | 1.00 | 1.00 | 1.00 | 1.00 | 1.00 |
| CIHR015094 | No | Yes | $2.08 \times 10^3$ | $1.12 \times 10^3$ | $3.62 \times 10^3$ | $1.47 \times 10^3$ | $4.42 \times 10^3$ |
| CIHR015434 | No | Yes | $2.71 \times 10^2$ | $1.10 \times 10^2$ | $5.04 \times 10^1$ | $1.52 \times 10^2$ | $1.42 \times 10^2$ |
| CIHR016024 | No | Yes | $2.69 \times 10^2$ | $3.41 \times 10^2$ | 1.00 | $3.60 \times 10^1$ | $4.24 \times 10^2$ |
| CIHR016624 | No | Yes | $7.33 \times 10^1$ | $5.24 \times 10^1$ | $2.57 \times 10^1$ | $4.62 \times 10^1$ | $8.38 \times 10^1$ |
| CIHR016979 | No | Yes | $2.80 \times 10^2$ | $2.25 \times 10^2$ | 1.00 | $4.60 \times 10^2$ | $1.12 \times 10^3$ |
| CIHR017127 | No | Yes | $7.31 \times 10^1$ | $3.50 \times 10^1$ | 1.00 | 7.84 | 1.00 |
| CIHR017305 | No | Yes | $2.57 \times 10^2$ | $1.41 \times 10^2$ | $3.73 \times 10^1$ | $1.47 \times 10^2$ | $1.44 \times 10^2$ |
| CIHR017724 | No | Yes | $2.57 \times 10^3$ | $1.26 \times 10^3$ | $3.68 \times 10^2$ | $7.59 \times 10^2$ | $5.56 \times 10^2$ |
| CIHR017894 | No | Yes | $8.78 \times 10^1$ | $8.06 \times 10^1$ | $3.92 \times 10^1$ | $5.59 \times 10^1$ | $4.22 \times 10^1$ |
| CIHR017990 | No | Yes | $3.74 \times 10^2$ | $1.46 \times 10^2$ | $4.10 \times 10^1$ | $1.17 \times 10^2$ | $6.43 \times 10^1$ |
| **Unvaccinated + any of (RBD ± S) no N** | | | | | | | |
| CIHR014238 | No | No | 1.00 | 1.00 | 1.00 | 1.00 | 1.00 |
| CIHR014491 | No | No | 1.00 | 1.00 | 1.00 | 1.00 | 1.00 |
| CIHR014632 | No | No | 1.00 | 1.00 | 1.00 | 1.00 | 1.00 |
| CIHR014664 | No | No | 1.00 | 1.00 | 1.00 | 1.00 | 1.00 |
| CIHR014926 | No | No | 1.00 | 1.00 | 1.00 | 1.00 | 1.00 |
| CIHR015079 | No | No | 1.00 | 1.00 | 1.00 | 1.00 | 1.00 |
| CIHR015475 | No | No | 1.00 | 1.00 | 1.00 | 1.00 | 1.00 |
| CIHR015843 | No | No | 1.00 | 1.00 | 1.00 | 1.00 | 1.00 |
| CIHR015948 | No | No | 1.00 | 1.00 | 1.00 | 1.00 | 1.00 |

**TABLE 3** (Continued)

| Specimen no. | Vaccinated | Anti-N | $ID_{50}$ | | | | |
|---|---|---|---|---|---|---|---|
| | | | Wild-type | Alpha | Beta | Gamma | Delta |
| CIHR016403 | No | No | 1.00 | 1.00 | 1.00 | 1.00 | 1.00 |
| CIHR016447 | No | No | 1.00 | 1.00 | 1.00 | 1.00 | 1.00 |
| CIHR016548 | No | No | 1.00 | 1.00 | 1.00 | 1.00 | 1.00 |
| CIHR016557 | No | No | 1.00 | 1.00 | 1.00 | 1.00 | 1.00 |
| CIHR016973 | No | No | 1.00 | 1.00 | 1.00 | 1.00 | 1.00 |
| CIHR017530 | No | No | 1.00 | 1.00 | 1.00 | 1.00 | 1.00 |
| CIHR017945 | No | No | 1.00 | 1.00 | 1.00 | 1.00 | 1.00 |
| CIHR018000 | No | No | 1.00 | 1.00 | 1.00 | 1.00 | 1.00 |
| CIHR018002 | No | No | 1.00 | 1.00 | 1.00 | 1.00 | 1.00 |
| CIHR018166 | No | No | 1.00 | 1.00 | 1.00 | 1.00 | 1.00 |
| CIHR018178 | No | No | 1.00 | 1.00 | 1.00 | 1.00 | 1.00 |

For the "vaccinated and anti-N" group (Fig. 4A), there was reduced neutralization of Beta VLPs compared to wild-type VLPs (Friedman statistic of 16; $P = 0.0025$, approximate; Dunn's multiple-comparisons test adjusted $P$ value of 0.0032) (Fig. 4A).

For the "vaccinated and no anti-N" group (Fig. 4B), there was a significant reduction in neutralization capacity (Friedman statistic of 54; $P < 0.0001$, approximate) of Alpha (Dunn's multiple-comparisons test; $P = 0.023$), Beta (Dunn's multiple-comparisons test; $P < 0.0001$), Gamma (Dunn's multiple-comparisons test; $P = 0.0002$), and Delta (Dunn's multiple-comparisons test; $P = 0.023$) compared to wild-type VLPs. Neutralization of Beta was also reduced compared to Alpha (Dunn's multiple-comparisons test; $P = 0.0032$) and Delta VLPs (Dunn's multiple-comparisons test; $P = 0.0032$) (Fig. 4B).

For the "Unvaccinated and anti-N" group (Fig. 4C), there was a significant reduction (Friedman statistic of 39; $P < 0.0001$) in Alpha (Dunn's multiple-comparisons test; $P = 0.027$), Beta (Dunn's multiple-comparisons test; $P < 0.0001$), and Gamma (Dunn's multiple-comparisons test; $P = 0.032$) compared to wild-type VOCs. Neutralization of Beta was also reduced compared to Alpha (Dunn's multiple-comparisons test; $P = 0.0037$), Gamma (Dunn's multiple-comparisons test; $P = 0.032$), and Delta VLPs (Dunn's multiple-comparisons test; $P = 0.012$) (Fig. 4C).

## DISCUSSION

This study evaluated the neutralizing capacity of stored plasma from January to March of 2021 but did not attempt to estimate seroprevalence in Canadian blood donors. A separate analysis of 16,931 specimens from April 13 to 30 indicated that anti-S seroprevalence in Canadian blood (Elecsys anti-SARS-CoV-2 S, Roche Diagnostics, Laval, PQ, Canada) was 3.2% (21). In this study, we oversampled 320 (7.1%) of 4,500 specimens with evidence of anti-S and/or ant-RBD. These selected specimens were randomized to select a subset for VLP assays (Fig. 1). We recently used a similar oversampling approach to identify candidate specimens for VLP and 50% plaque reduction/neutralization titer ($PRNT_{50}$) studies during the first wave of the pandemic (6).

Our group has previously shown that the identification of anti-S, anti-RBD, or anti-N in blood donor specimens is not sufficient to predict the neutralizing antibody capacity of plasma against wild-type or variant SARS-CoV-2 (5, 6). It is also clear that the neutralizing capacity of plasma can vary widely in unvaccinated individuals (5, 7), even in the presence of anti-S and/or anti-RBD antibodies (6). Both anti-S and anti-RBD assays used in this study have been previously described as being highly specific (e.g., >98%) (2). Therefore, it is possible that this group of donors represents donors who failed to mount a neutralizing antibody response to SARS-CoV-2 (7) or have now lost their neutralizing antibodies as well as anti-N (7, 22). Within unvaccinated repeat routine Canadian blood donors, we have previously noted a trend to waning anti-N over time with the Abbott anti-N immunoassay and seroreversion of donors over time, with an adjusted rate of decline of $-0.06$ U/week (95% confidence interval of $-0.08$ to $-0.05$) (22). Within Canadian convalescent plasma donors, we also noted a failure for some individuals to mount a strong neutralizing antibody response against wild-type SARS-

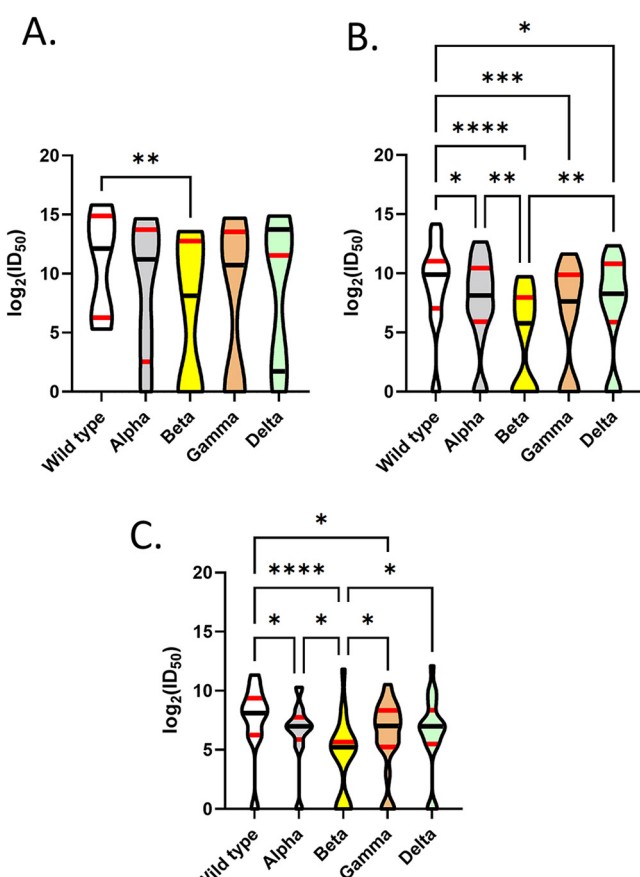

**FIG 4** Differences in VLP neutralization between wild-type, Alpha, Beta, Gamma, and Delta VLPs within vaccinated and unvaccinated groups. Comparison of $\log_2$ ($ID_{50}$) against wild-type and variant VLPs within vaccinated and unvaccinated groups. Data are represented as violin plots with medians (black horizontal lines), 25th percentiles (lower red horizontal lines) and 75th percentiles (upper red horizontal lines). Truncated violin plots range from minimum to maximum data points. Pairwise comparisons were undertaken between VOCs (A to C). (A) "Vaccinated and anti-N." A significant reduction in the neutralization of Beta VLPs to wild-type constructs was identified in this group. (B) "Vaccinated and no anti-N." A significant reduction was identified in the neutralization of Alpha, Beta, Gamma, and Delta VLPs compared to wild-type VLPs. A significant reduction was also identified in the neutralization of Beta compared to Alpha and Delta VLPs. (C) "Unvaccinated and anti-N." A significant reduction was identified in the neutralization of Alpha, Beta, and Gamma VLPs compared to wild-type VLPs. A significant reduction was also identified in the neutralization of Beta compared to Alpha, Gamma, and Delta VLPs. For the "unvaccinated and no anti-N" group, no measurable neutralization of wild-type or Alpha, Beta, Gamma, and Delta VLPs was identified in this group, and this group is not visible on this figure. Asterisks represent significant differences in neutralizing capacity of plasma against wild-type and variant VLPs as determined by Dunn's multiple-comparison testing.

CoV-2 soon after infection. Within that convalescent plasma donor population, we also noted a decrease in neutralizing antibody activity, as measured by a wild-type $PRNT_{50}$ test over a period of 3 to 4 months after infection (7). We cannot rule out that some donors within this group may be generating anti-S and anti-RBD against other coronaviruses (23) or may be false-positive anti-S or anti-RBD due to assay characteristics (2).

Within the "unvaccinated and anti-N"-positive group (Fig. 4C), we also found a reduced neutralizing capacity against Beta VLPs compared to wild-type VLPs as well as Alpha, Gamma, and Delta VLPs. Neutralization of Beta VLPs was also reduced compared to Alpha, Gamma, and Delta VLPs. In a previous pilot study in unvaccinated routine blood donors with prior evidence of infection (e.g., anti-S and/or anti-RBD), we noted a variability in plasma specimens' ability to neutralize Beta VLPs and live virus (6). Similarly, a Strasbourg cohort of convalescent plasma collected prior to the emergence of the VOCs also showed a reduction of neutralizing capacity against Beta (24, 25). Convalescent plasma collected from patients in the United States during the spring of 2020 also showed a reduced

neutralizing capacity against Beta (26). In a recent meta-analysis by Chen et al., serum from previously infected individuals had the greatest decrease in Beta (4.1-fold) compared to wild-type neutralization in live virus neutralization studies. This was then followed by decreased Delta (3.2-fold), Gamma (1.8-fold), and Alpha neutralization (1.4-fold) (27). We note that fold changes described in live virus studies may not translate to VLP studies. This variability to neutralize SARS-CoV-2 VOCs was previously described by our group using plasma collected in April and May of 2020 (6).

Donor plasma in the "vaccinated and no anti-N" group (Fig. 4B) tended to have peak and median neutralization to wild type that was similar to the "vaccinated with anti-N" group (Fig. 4A). However, compared to wild-type VLPs, there was reduced neutralization of Alpha, Beta, Gamma, and Delta VLPs in this group. Neutralization of Beta VLPs was also reduced compared to Alpha and Delta VLPs. This group may represent donors who were vaccinated only or donors who were infected and vaccinated and now have a waned anti-N (2, 3, 22). A pooled meta-analysis has recently described heterogeneity in studies assessing the neutralizing capacity of vaccinated individuals against VOCs, which may depend on a variety of factors, including agents (e.g., live virus versus pseudovirus) used in experiments (27). However, that meta-analysis did note that geometric mean titers were lowest against Beta in a variety of studies focused on vaccinated individuals (27). It is important to highlight that there was no absolute loss of neutralizing capacity in plasma from this group against Alpha, Beta, Gamma, and Delta.

The smallest group of specimens available in this study was from the "vaccinated and anti-N" group ($n = 5$). We know from previous studies that less than 5% of Canadian blood donors exhibited evidence of infection by September of 2020 (2). From a large unpublished seroprevalence study of Canadian blood donors, we also know that as of May 2021, that seroprevalence in this population had only reached 4% (21). Furthermore, as of late March 2021, only 9.18% of the Canadian population had received at least one dose of vaccine (13). This group would also most likely represent vaccinated blood donors who also had a history of natural SARS-CoV-2 infection and who had not yet lost anti-N (22). The small numbers of plasma specimens within this group exhibited reduced neutralizing capacity against Beta VLPs compared to wild-type VLPs. Unlike the "vaccinated and no anti-N" group, reduced neutralization capacity was not significant against other variant VLPs and might be a product of cross-variant neutralizing capacity in vaccinated and previously infected individuals (28, 29). Other trends in variant neutralization may have not been evident due to the small number of specimens in this group.

Vaccination against SARS-CoV-2 provides substantial protection against severe outcomes even with an ecological niche now dominated by Delta (30, 31). Previous work has suggested that fully vaccinated individuals are less likely to be reinfected than previously infected individuals (32). It is also clear that vaccinated individuals can still mount effective, albeit reduced, neutralization against VOCs (27, 33). Two factors may be at play in this study, (i) absolute neutralizing titers against wild-type and variant SARS-CoV-2 and (ii) breadth of antibody protection against one or more strains of SARS-CoV-2. In this study, wild-type and variant neutralizing capacity was higher in vaccinated donor specimens than in nonvaccinated donor specimens who exhibited serological evidence for SARS-CoV-2 infection (Fig. S1A to E in the supplemental material). This was evidenced by the higher absolute peak and median neutralization values for vaccinated individuals as well as lack of any neutralizing capacity in a subset of unvaccinated donors (unvaccinated and no anti-N) (Fig. 4A to D; Table S2; Fig. S1A to E). Higher absolute neutralizing capacity may be an important defense mechanism in infections involving exceedingly high viral titers of Delta virus (34, 35). It is also important to note that plasma from vaccinated donors displayed a greater breadth of protection against strains of SARS-CoV-2 than did plasma from unvaccinated donors.

As previously mentioned, during this time period, less than 2% of the Canadian population had received a second dose of vaccine (13). A recent preprint suggests that a single dose of an mRNA (Pfizer-BioNTech [BNT162b2] or mRNA-1273 [Moderna]) vaccine produced neutralization titers like those measured in convalescent individuals. The study was

also not able to determine which vaccine was received by donors. A recent meta-analysis of immunogenicity studies suggests that mRNA and vector vaccines stimulate measurable antibody responses to SARS-CoV-2 wild type and VOCs (36). In contrast, Ad26.COV2.S (Janssen) yielded lower antibody concentrations and frequently negative neutralization titers (37). Geometric mean titer (GMT) values in a proxy assay for wild-type neutralization have been shown to double after the second dose of ChAdOx1-nCOV (38).

There are several key limitations to this study that have not already been addressed. First, this study involved a relatively small number of specimens collected from Canadian blood donors collected in a low-seroprevalence setting and prior to large-scale national vaccine coverage (2, 13, 21). The study was not able to identify if donors had a laboratory or clinical diagnosis of SARS-CoV-2 and when infection occurred. This therefore prevents us from developing an understanding of the impact of waning neutralizing humoral immunity in this study population (7). One drawback to our study is that for operational and ethics approval reasons, we were unable to capture information on vaccine type and if a second dose of vaccine had been administered. We were only able to assesses donors for the receipt of a SARS-CoV-2 vaccine in the prior 3 months and not the timing of the vaccine (39). These experiments utilized VLPs and not PRNT$_{50}$ assays using live wild-type or variant SARS-CoV-2. We have not seen concerning trends in extrapolating conclusions from VLP-based experiments in our prior work (20) nor in other studies but intend to explore these differences soon (6, 27). Finally, this study also does not account for cell-mediated immunity as well as the development of sustained B cell immunity in vaccinated and previously infected blood donors (40–42).

In the first 3 months of 2021, Canadian blood donors had various levels of humoral protection against wild-type and variant SARS-CoV-2. The highest absolute levels of neutralizing capacity were in vaccinated blood donors, albeit with reduced neutralization capacity against SARS-CoV-2 VOCs. One concern highlighted in this study was that unvaccinated blood donors not only had reduced neutralizing capacity against Beta but, in the absence of anti-N, also had no measurable neutralization capacity against wild-type and variant SARS-CoV-2. Therefore, the manuscript highlights the importance of vaccination in populations with low levels of seroprevalence to SARS-Co-V2, even in individuals with prior SARS-CoV-2 infection.

## MATERIALS AND METHODS

**Ethical considerations.** Ethics board clearance was provided by Canadian Blood Services, the University of Alberta, and the Sinai Health, Toronto (Lunenfeld-Tanenbaum Research Institute).

**CIHR correlates of immunity study participants and samples.** Canadian Blood Services collects blood donations in all provinces except Quebec, with collection sites concentrated in large and small cities. Blood donors (≥17 years age) meet rigorous health selection criteria, and blood donations are used to manufacture products for transfusion. With each donation, an additional EDTA plasma retention sample is also collected alongside each donation (43).

This study used a repeated cross-sectional design. A random cross-sectional sampling of all available Canadian Blood Services retention specimens ($n$ = 1,500/month) was selected monthly for January, February, and March 2021.

**Collection of SARS-CoV-2 vaccination history in donors.** At the time of donation, all donors were asked if they recently had a vaccination within the prior 3 months. Specifically, information on SARS-CoV-2 vaccination was routinely captured on the record of donation as standard practice by this blood operator. Due to ethics approval and operational issues, information on exact timing, vaccine producer, and dosing schedule (e.g., first or second dose) was not collected. Data from provincial vaccine databases could also not be linked to the records of donation. These data were stored in the donor database.

Samples were anonymized. Data, including donation date, birth year, sex, collection site, and residential Forward Sortation Area (FSA; first three characters of postal code), were extracted from the donor database. Plasma specimens were aliquoted at Canadian Blood Services. One aliquot (250 μL) was stored at –80°C for the remainder of the study.

**Enzyme-linked immunosorbent assays for detecting IgG.** IgG analysis was undertaken at the Lunenfeld-Tanenbaum Research Institute, Sinai Health, on all retained specimens. Assays were anti-spike (anti-S), anti-receptor-binding domain (anti-RBD), and anti-nucleocapsid protein (anti-N), as previously described (19). Ratio-converted enzyme-linked immunosorbent assay (ELISA) reads were undertaken as previously described (5, 20), and cutoffs (positive) of each of the targets were ≥0.396 for N, ≥0.186 for RBD, and ≥0.190 for S (2).

**Abbott architect SARSCoV-2 IgG test.** Plasma samples were also tested with the Abbott Architect SARS-CoV-2 IgG test (Abbott Laboratories, USA), which detects anti-N IgG antibodies, as directed by the manufacturer, using an antibody index (AI) cutoff of 1.4 (2, 3).

**Definition of an anti-N-positive specimen.** A specimen was defined as anti-N positive if it had one or more anti-N-positive signals from either the Abbott or the Lunenfeld-Tanenbaum Research Institute, Sinai Health, assays.

**Selection of specimens for neutralizing analysis.** As previously described, a tiered testing approach of specimens with any potential evidence of a signal for anti-S or anti-RBD (with or without anti-N) was used to select specimens for further analysis by neutralization methods (4, 5). Specimens were then randomized with replacement, as previously described (2).

**Spike-pseudotyped VLP neutralization assay.** Pseudovirus generation was performed in HEK293TN cells (a cell line optimized for lentiviral particle production; Systems Biosciences; RRID:CVCL_UL49), as previously described (5, 20), with minor modifications. Briefly, VLPs were generated from cotransfection with jetPRIME (Polyplus Inc.) of HEK293TN cells (in 10% fetal bovine serum [FBS] and 1% penicillin/streptomycin [Pen/Strep] in Dulbecco's modified Eagle's medium [DMEM]) with (i) viral packaging (psPAX2, Addgene), (ii) ZsGreen and luciferase reporter (pHAGE-CMV-Luc2-IRES-ZsGreen-W, kindly provided by Jesse Bloom), and (iii) spike protein constructs. Spike constructs were wild-type (a sequence based on Wuhan-Hu-1 but bearing the D614G mutation) and the following VOCs: Alpha (B.1.1.7), Beta (B.1.351), Gamma (P.1), and Delta (B.1.617.2); all were kindly provided by W. Rod Hardy of CoVaRR-Net. After 8 h of transfection, the medium was replaced by DMEM containing 5% heat-inactivated FBS and 1% Pen/Strep, and the cells were incubated for 16 h at 37°C and 5% $CO_2$; they were then transferred to 33°C and 5% $CO_2$ for an additional 24 h for VLP production. At 48 h after transfection, the supernatant was collected, centrifuged at 500 $\times$ $g$ for 5 min at room temperature, filtered through a 0.45-$\mu$m filter, and frozen in aliquots at −80°C. The pseudovirus particles were used at a dilution resulting in greater than 1,000 relative luciferase units (RLUs) over control (1:20 to 1:250 dilution of virus stock, depending on the virus titers of each variant). For the neutralization assay, human sera were heat inactivated (56°C, 1 h) and diluted at 1:20 in assay medium and then serially diluted by 2.5-fold over 7 dilutions before incubation with diluted pseudovirus at a 1:1 ratio for 1 h at 37°C. The virus and serum mixture was then transferred onto HEK293T-ACE2/TMPRSS2 cells (maintained in 10% heat-inactivated FBS and 1% Pen/Strep in DMEM), a stable cell population resulting from the serial transduction (with a multiplicity of infection [MOI] of <1) first of angiotensin-converting enzyme 2 (ACE2) (in pLenti CMV Puro DEST, Addgene, 17452) and selection with puromycin (1 $\mu$g/mL) then with TMPRSS2 (in pLenti CMV Hygro DEST, Addgene, 17454) and selection with hygromycin (150 $\mu$g/mL). Both the HEK293TN and HEK293T-ACE2/TMPRSS2 cells were maintained in indicated DMEM medium and passaged before they reached 85% confluence; cells were not kept in culture for more than 25 passages. The infected cells were lysed after 48 h using BrightGlo luciferase reagent (Promega, Madison, WI) and read with a PerkinElmer EnVision instrument. Unless otherwise specified, the 50% neutralization titers ($ID_{50}$) were generated in GraphPad Prism 9, using a nonlinear regression algorithm (log [inhibitor] versus normalized response – variable slope). Data were further cleaned, analyzed, and visualized using Python (v. 3.9.7) and Seaborn (0.11.2). The assay performer was blinded to the patient sample groups.

**Data storage and statistical analysis.** Blood donor data were stored on a Microsoft Excel (Redmond, WA, USA) spreadsheet. Data were analyzed as described in Results using GraphPad Prism (9.2.0, GraphPad Software, Inc, San Diego, CA, USA). Data analyses undertaken using this statistical program included descriptive statistics, Fisher's exact test (two sided), and a Mann-Whitney U test. Friedman statistics with Dunn's multiple-comparison tests were used when there was analysis of repeated measures within a group, such as when comparing wild-type, Alpha, Beta, Gamma, and Delta VLP neutralization within a group of plasma (e.g., "vaccinated and anti-N"). Kruskal-Wallis with Dunn's multiple-comparisons statistics were used when repeated measures were not used and neutralization of a specific VOC (e.g., Alpha) was compared across plasma from different groups (e.g., tested "vaccinated and anti-N" versus "vaccinated and no anti-N" versus "not vaccinated and anti-N" versus "not-vaccinated and no anti-N" plasma against Alpha VLPs). Graphing of data used GraphPad Prism.

## SUPPLEMENTAL MATERIAL

Supplemental material is available online only.

**SUPPLEMENTAL FILE 1**, PDF file, 0.2 MB.

## ACKNOWLEDGMENTS

We are grateful to W. Rod Hardy for the spike constructs and Jesse Bloom for VLP packaging reagents. We thank Craig Jenkins and Valerie Conrod for assistance with testing and coordination. We are also grateful to Canadian Blood Services staff and leadership for support for this project.

The following authors have no conflicts of interest: S.F.O., Q.H., R.S., K.T.A., B.R., K.C., and Q.-L.Y. S.J.D. has functioned as a content expert for respiratory viruses for Johnson & Johnson (Janssen). A.-C.G. receives research funding from Providence Therapeutics Holdings, Inc., for other projects. A.-C.G. is the Canada Research Chair (Tier 1) in Functional Proteomics and pillar lead for CoVaRR-Net.

S.J.D., S.F.O., and A.-C.G. received funding through the Canadian Institutes of Health Research (CIHR; VR2-172723) and Alberta Innovates (G2020000360 S.J.D.). A.-C.G. also received funding through the Krembil Foundation to the Sinai Health System Foundation, Ontario Together, and CIHR (VR1-172711, with supplement from the

COVID-19 Immunity Task Force). The robotics equipment used for the ELISA assays is housed in the Network Biology Collaborative Centre at the Lunenfeld-Tanenbaum Research Institute (A.-C.G.), a facility supported by Canada Foundation for Innovation funding by the Ontario Government and by Genome Canada and Ontario Genomics (OGI-139). The development of the spike protein constructs and VLP assays was supported by the CIHR operating grant to the Coronavirus Variants of Concern Rapid Response Network (CoVaRR-Net) to A.-C.G. Commercial Abbott Architect SARS-Cov-2 IgG assay kit costs were partially supported by Abbott Laboratories, Abbott Park, IL. Abbott analyzers used at Canadian Blood Services were provided by the COVID-19 Immunity Task Force (CITF). The funders had no role in study design, data collection and analysis, decision to publish, or preparation of the manuscript.

Q.H., R.S., K.T.A., and K.C. developed the methodology. Q.H., R.S., and B.R. performed the investigations. A.-C.G., S.J.D., and S.F.O. acquired funding. A.-C.G., S.J.D., and S.F.O. supervised the project. S.J.D. drafted the manuscript. S.F.O., Q.-L.Y., and S.J.D. performed data collation and analysis.

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
