## [Reviewer comments · Microbiology Spectrum]

Microbiology Spectrum

SARS-CoV-2 virus-like particle neutralizing capacity in blood donors depends on serological profile and donor declared SARS-CoV-2 vaccination history.

Steven Drews, Queenie Hu, Reuben Samson, Kento Abe, Bhavisha Rathod, Karen Colwill, Anne-Claude Gingras, Qi-Long Yi, and Sheila O'Brien

Corresponding Author(s): Steven Drews, Canadian Blood Services

Review Timeline:

Submission Date:	November 12, 2021
Editorial Decision:	December 18, 2021
Revision Received:	January 5, 2022
Accepted:	January 11, 2022

Editor: Heba Mostafa

Reviewer(s): The reviewers have opted to remain anonymous.

Transaction Report:

DOI: <https://doi.org/10.1128/spectrum.02262-21>

December 18, 2021

Dr. Steven J Drews
Canadian Blood Services
8249 114 St NW
Edmonton, Alberta T6G 2R8
Canada

Re: Spectrum02262-21 (SARS-CoV-2 virus-like particle neutralizing capacity in blood donors depends on serological profile and donor declared SARS-CoV-2 vaccination history.)

Dear Dr. Steven J Drews:

Link Not Available

Sincerely,

Heba Mostafa

Journals Department
Reviewer comments:

Reviewer #1 (Comments for the Author):

The authors describe a serosurvey by the Canadian blood services looking at the ability of seropositive samples to neutralize different SARS-CoV-2 lineages. This study was done in a time where vaccination rates were low in Canada, and the results are valuable for investigating the use of sera as possible therapeutics. The study was carefully crafted and the limitations clearly noted. No major issues were found. The only minor concern is the length of the manuscript, which slightly exceeds the word limit for the journal. Given the complexity of the data, I would advocate for an exception. Formatting can be addressed if the manuscript moves forward, and I would also suggest putting tables into the supplemental material to focus on the Figure which summarize the data.

Reviewer #2 (Comments for the Author):

This study analyzed 4500 Canadian blood donors with a known vaccination status during the first 3 months of 2021 at the

beginning of vaccination rollout for presence of anti-SarsCov2 antibodies with anti-N, anti-S and anti-RBD reactivity. A few samples were then characterized for the ability to neutralize Sars-Cov2 wild-type and variant of concern strains. Vaccinated donors showed elevated Sars-Cov2 antibodies titers with strong neutralizing capacity against wild-type and variants (slightly reduced against the beta variant). Interestingly, at the time of sample collection, half of the unvaccinated donors had anti-Sars-Cov2 antibodies with acceptable neutralizing capacity.

While it provides useful information, the manuscript is difficult to read. All sections are long and detailed, and should be more concise highlighting the important findings. Some efforts should be put on data presentation and description of the results. Particularly, the same data are presented several times in different ways (Figures, Table, Heatmap) and according to various statistical comparisons. The authors should simplify selecting for example one mean of presentation. The discussion brings a good perspective but can be shortened. The limitations of the study are well explained. The main conclusions of the study should be highlighted, especially in the abstract.

Important comments:

1. The results section can be improved, especially when describing Fig. 4 and 5. The same phrases ("neutralization was elevated/reduced...", ...) are repeated many times. The authors should describe their findings in a more concise manner.
2. Tables 4 and 5 present the same data (untransformed and transformed). Only one should be shown. Similarly, the summary data tables (Table 5 and 6) are unnecessary as the data are already shown in Fig. 4, 5, 6. The heatmap is also a different way to present the same data and does not add information to the manuscript. Those additional supportive figure/table(s) can be added to Supplemental Information.
3. It is not clear why the authors chose multiple different statistical test to describe similar data. For example, the data in Fig. 4 and 5 are the same but presented in a different order and the authors used Friedman (Fig. 4) and Kruskal-Wallis (Fig. 5). A few lines giving a rationale of the choice of the statistical test should be explicit?

Other specific comments:

4. The age data corresponding to L155-157 (median, range) should be added in Table 1.
5. The authors should show the cut-off directly on the graphs in Fig. 2 and 3 (dashed line) to make it easier to read.
6. The graph in Panel D in Fig. 4 is empty and cannot be displayed this way, at least not without a positive result on the graph. This result is also shown in a table and in the next figure, and those can be referred to.
7. L173-178: to which figure/table do those lines refer to?
8. Please add a paragraph in the method section describing cell origin and culture conditions?
9. L421-425 and L433-435 in the method section are duplicated and one should be removed.
10. Fig. 6 needs a better resolution or resizing.
11. Please double check the text for typos/errors/punctuations.
12. Please also check that the references are complete and without unnecessary information.

Staff Comments:

Preparing Revision Guidelines

Please return the manuscript within 60 days; if you cannot complete the modification within this time period, please contact me. If you do not wish to modify the manuscript and prefer to submit it to another journal, please notify me of your decision immediately so that the manuscript may be formally withdrawn from consideration by Microbiology Spectrum.

To the Editor,
Microbiology Spectrum

Re: Spectrum02262-21 (SARS-CoV-2 virus-like particle neutralizing capacity in blood donors depends on serological profile and donor declared SARS-CoV-2 vaccination history.)

Thank you for reviewing our manuscript. We have responded to reviewer comments below using a point-by-point approach. Edits in the manuscript are marked up in the text.

Reviewer #1 (Comments for the Author):

The authors describe a serosurvey by the Canadian blood services looking at the ability of seropositive samples to neutralize different SARS-CoV-2 lineages. This study was done in a time where vaccination rates were low in Canada, and the results are valuable for investigating the use of sera as possible therapeutics. The study was carefully crafted and the limitations clearly noted. No major issues were found. The only minor concern is the length of the manuscript, which slightly exceeds the word limit for the journal. Given the complexity of the data, I would advocate for an exception. Formatting can be addressed if the manuscript moves forward, and I would also suggest putting tables into the supplemental material to focus on the Figure which summarize the data.

Thank you for the comments, we have addressed the length issue by primarily responding to Reviewer #2. In general, we made the results section more concise and have moved some figures and tables to supplemental files. Other redundant data has been removed.

Reviewer #2 (Comments for the Author):

While it provides useful information, the manuscript is difficult to read. All sections are long and detailed, and should be more concise highlighting the important findings. Some efforts should be put on data presentation and description of the results. Particularly, the same data are presented several times in different ways (Figures, Table, Heatmap) and according to various statistical comparisons. The authors should simplify selecting for example one mean of presentation. The discussion brings a good perspective but can be shortened. The limitations of the study are well explained. The main conclusions of the study should be highlighted, especially in the abstract.

Thank you for the comments, we have addressed the length issue by reducing text and moving some figures and tables to supplemental files. Some redundant data has been removed from the text.

Important comments:

1. The results section can be improved, especially when describing Fig. 4 and 5. The same phrases ("neutralization was elevated/reduced...", ...) are repeated many times. The authors should describe their findings in a more concise manner.

We have reduced the text and made it more concise.

2. Tables 4 and 5 present the same data (untransformed and transformed). Only one should be shown.

Similarly, the summary data tables (Table 5 and 6) are unnecessary as the data are already shown in Fig. 4, 5, 6.

We have moved Tables 4,5 and 6 to supplementary files.

The heatmap is also a different way to present the same data and does not add information to the manuscript.

We have deleted the heat map.

Those additional supportive figure/table(s) can be added to Supplemental Information.

3. It is not clear why the authors chose multiple different statistical test to describe similar data. For example, the data in Fig. 4 and 5 are the same but presented in a different order and the authors used Friedman (Fig. 4) and Kruskal-Wallis (Fig. 5). A few lines giving a rationale of the choice of the statistical test should be explicit?

Friedman statistics were used when there was analysis of repeated measures within a group. For example when comparing Wild type, Alpha, Beta, Gamma and Delta VLP neutralization within a group of plasma (e.g. "vaccinated and anti-N"). Kruskal-Wallis statistics were used when repeated measures were not used and different groups (e.g. tested "vaccinated and anti-N" vs "vaccinated and no anti-N" vs "not vaccinated and anti-N" vs "not-vaccinated and no anti-N" plasma" were compared for neutralization against specific VLPs (e.g. Alpha).

Our primary description of the data will be with Figure 4. Figures 5 and 6 will be moved to supplemental files

Other specific comments:

4. The age data corresponding to L155-157 (median, range) should be added in Table 1.
Age data was added to Table 1.

5. The authors should show the cut-off directly on the graphs in Fig. 2 and 3 (dashed line) to make it easier to read.

Cut-offs for each of the assays were drawn onto and Fig. 2 and Fig.3.

6. The graph in Panel D in Fig. 4 is empty and cannot be displayed this way, at least not without a positive result on the graph. This result is also shown in a table and in the next figure, and those can be referred to.

Panel D had been removed from Figure 4.

7. L173-178: to which figure/table do those lines refer to?

We have deleted this text as it is redundant to later text.

8. Please add a paragraph in the method section describing cell origin and culture conditions?

We have added information in a restructured text.

9. L421-425 and L433-435 in the method section are duplicated and one should be removed.

The duplication was removed.

10. Fig. 6 needs a better resolution or resizing.

We have removed Fig. 6 as it was redundant.

11. Please double check the text for typos/errors/punctuations.

We have analyzed our manuscript text for conciseness, spelling, grammar, clarity, conciseness and formality using the Word "editor." We have made changes where appropriate.

12. Please also check that the references are complete and without unnecessary information.

We have reviewed references and added doi and web links where full publication information is not yet available on Pubmed.

Regards

Steven J. Drews PhD FCCM D(ABMM)

Canadian Blood Services, 8249 114 St NW

Edmonton, AB T6G 2R8, email: steven.drews@blood.ca, phone: 780-702-8639

January 11, 2022

Dr. Steven J Drews
Canadian Blood Services
8249 114 St NW
Edmonton, Alberta T6G 2R8
Canada

Re: Spectrum02262-21R1 (SARS-CoV-2 virus-like particle neutralizing capacity in blood donors depends on serological profile and donor declared SARS-CoV-2 vaccination history.)

Dear Dr. Steven J Drews:

Your manuscript has been accepted, and I am forwarding it to the ASM Journals Department for publication. You will be notified when your proofs are ready to be viewed.

Sincerely,

Heba Mostafa
Editor, Microbiology Spectrum
